

# Accurately calibrated XRF-CS record of Ti/Al reveals Early Pleistocene aridity/humidity variability over North Africa and its close relationship to low-latitude insolation

Rick Hennekam[1], Katharine M. Grant[2], Eelco J. Rohling[2,3], Rik Tjallingii[4], David Heslop[2], Andrew P. Roberts[2], Lucas J. Lourens[5], Gert-Jan Reichart[1,5]

[1]Department of Ocean Systems, NIOZ Royal Netherlands Institute for Sea Research, P.O. Box 59, 1790 AB Den Burg, Texel, The Netherlands
[2]Research School of Earth Sciences, Australian National University, Canberra, ACT 2602 Australia
[3]Ocean and Earth Science, University of Southampton, National Oceanography Centre, Southampton, SO14 3ZH, United Kingdom
[4]GFZ German Research Centre for Geosciences, Section 5.2 – Climate Dynamics and Landscape Evolution, D-14473 Potsdam, Germany
[5]Department of Earth Sciences, Faculty of Geosciences, Utrecht University, P.O. Box 80.121, 3508 TA Utrecht, The Netherlands

*Correspondence to*: Rick Hennekam (rick.hennekam@nioz.nl)

**Abstract.** In eastern Mediterranean Sea sediments, the titanium to aluminum ratio (Ti/Al) captures relative variability in eolian to riverine derived material, and predominantly integrates climate signals over the Saharan and Sahel regions. Long Ti/Al time series can, therefore, provide valuable records of North African humidity/aridity changes. X-ray fluorescence core scanning (XRF-CS) can generate near-continuous Ti/Al records with relatively modest effort and in an acceptable amount of time, provided that accurate Ti/Al values are acquired. Calibration of the raw XRF-CS data to those of established analytical methods is an important pathway to obtain this required accuracy. We assess how to obtain reliable XRF-CS Ti/Al calibration by using different sets of calibration reference samples for a long sediment record from ODP Site 967 (eastern Mediterranean). The accuracy of reference concentrations and the number of reference samples are important components for reliable calibration. The acquired continuous Ti/Al record allows detailed time-series analysis over the past 3 Ma. A near-direct control of low-latitude insolation on the timing and amplitude of North African aridity/humidity is observed from 3 to ~1.2 Ma. It is evident from our Ti/Al record that the most arid North African intervals (i.e., with longest period and highest amplitude) occur after the mid-Pleistocene transition (MPT; ~1.2-0.7 Ma). Concurrently, correlation between North African aridity/humidity (Ti/Al) and higher latitude climate signals (ice-volume variability) increases around the MPT. These findings support the growing consensus that African climate became more sensitive to remote high-latitude climate when a threshold ice volume was reached during the MPT.



## 1 Introduction

Continuous Pliocene-Pleistocene records that capture variability in North African aridity and humidity are sparse. Yet, such records are crucial for understanding the link between insolation, high latitude climate, and low latitude climate in Africa

during the Plio-Pleistocene (deMenocal, 1995; Trauth et al., 2009; 2021), when Northern Hemisphere glaciation intensified (e.g., Shackleton et al., 1984; Mudelsee and Raymo, 2005; Rohling et al., 2021). Moreover, such records are essential for providing climatic context to contemporaneous hominin evolutionary events and out-of-Africa dispersals (Blanchet et al., 2021; deMenocal, 2011; Donges et al., 2011; Kaboth-Bahr et al., 2021; Larrasoaña et al., 2013; Larrasoaña, 2021; Maslin et al., 2014; Rohling et al., 2013; Trauth et al., 2021). Until now, long (>Myr) and continuous North African records have

mainly focused on dust fluxes from subtropical West Africa (Ocean Drilling Program (ODP) Site 659; Tiedemann et al., 1994), the Arabian Peninsula and Horn of Africa (ODP Sites 721/722; Bloemendal and deMenocal, 1989; deMenocal, 1995, 2004), and (sub-)Saharan North Africa (ODP Site 967; Larrasoaña et al., 2003; Grant et al., 2022) (Fig. 1). Although dust flux reconstructions tend to track large-scale continental aridity changes, they specifically relate to dust source extent, ablation potential, and eolian transportation. Hence, they do not necessarily track the intensity and extent of wet episodes. To

date, the only continuous high-resolution records that capture relative humidity/aridity changes in North Africa throughout Pleistocene are from ODP Site 967: i) a "wet-dry index" (Grant et al., 2017), which combines a monsoon run-off proxy with an existing eolian record, and ii) a recently derived titanium (Ti) to aluminum (Al) record spanning the last 5 Ma (Grant et al., 2022).

The Ti/Al ratio in bulk sediments from the eastern Mediterranean Sea has been found to track aridity versus humidity

variability in the Sahel and Saharan regions (Lourens et al., 2001; Wehausen and Brumsack, 2000). This proxy relies on the assumption that the eolian fraction that reaches the Levantine basin is enriched in Ti over Al, while the fluvial fraction comprises mainly clays, such as smectites, that are relatively enriched in Al over Ti (Lourens et al., 2001; Wehausen and Brumsack, 2000). Dust fluxes from North Africa toward the eastern Mediterranean Sea were generally enhanced during arid episodes in the source regions (Larrasoaña et al., 2003; Trauth et al., 2009), while riverine sediment fluxes increase during

intervals with enhanced humidity and related African continental runoff (Williams et al., 2006). The ODP Site 967 Ti/Al record is, thus, interpreted as a North African aridity/humidity indicator, with high values representing arid intervals and low values corresponding to more humid periods (Lourens et al., 2001).

The Ti/Al records from ODP Site 967 and neighboring Site 968 have been used previously to deduce monsoon variability over North Africa at 0-1.2 Ma and 2.3-3.2 Ma (De Boer et al., 2021; Konijnendijk et al., 2014; 2015; Lourens et al., 2001).

These studies showed cyclic North African climate variability in tune with insolation during the 2.3-3.2 Ma interval (De Boer et al., 2021; Lourens et al., 2001), which persisted during the 0-1.2 Ma interval, albeit with considerably more lag (up to 3-4 kyr) between precession minima/insolation maxima and North African climate (Konijnendijk et al., 2014; 2015) depending on sea-level and monsoon changes (Grant et al., 2016). These records were produced using wavelength dispersive X-ray fluorescence (WD-XRF) analyses on molten glass beads, which is an established accurate and precise geochemical



approach (Wehausen and Brumsack, 2000), yet also relatively costly, destructive, and time consuming if a high sampling

resolution is required (Wilhelms-Dick et al., 2012). Future work would benefit from relatively rapid, cost-efficient, and non-

destructive approaches to produce reliable Ti/Al records, such as XRF core scanning (XRF-CS) (Croudace and Rothwell,

2015; Jansen et al., 1998). Grant et al. (2017) generated geochemical records using XRF-CS on ODP Site 967 sediments,

which were calibrated using samples measured by energy-dispersive XRF (ED-XRF) on sediment powders. This resulted in

XRF-CS-based elemental concentration and ratio profiles that generally agree well with existing quantitative geochemical

data from this core. However, the XRF-CS Ti/Al ratio revealed offsets with conventional measurements (Grant et al., 2017),

which so far remain unexplained. Grant et al. (2022) have now extended the ODP Site 967 XRF-CS records back to 5 Ma,

and successfully recalibrated the entire 0-5 Ma interval using a larger number of reference samples, all of which were

obtained by WD-XRF on glass beads. It therefore appears that the number of reference samples and/or the analytical

technique (i.e., accuracy of the analyses) are critical for accurate XRF-CS calibration.

Here we first assess how to obtain reliable calibrated Ti/Al XRF-CS profiles, with application to the ODP Site 967

record, with relevance to more general use of XRF-CS Ti/Al and potentially also other element ratios. We focus on

calibration sample selection and the accuracy of calibration concentrations, which are used to convert qualitative Ti/Al

values (counts/counts) from XRF-CS into quantitative Ti/Al values (ppm/ppm) (e.g., Weltje et al., 2015). We use the

extensive (N = 1060) set of glass-bead WD-XRF measurements of Konijnendijk et al. (2014; 2015) instead of those

produced with polarized ED-XRF on sediment powder samples (N = 40) of Grant et al. (2017), because the glass-bead WD-

XRF technique is regarded to be more precise and accurate than ED-XRF on sediment powders (Zhan, 2005). The large

sample set of Konijnendijk et al. (2014; 2015) also enables investigation of the number of samples required for accurate

XRF-CS calibration. Finally, we use the obtained Ti/Al record to study the relationship of North African climate to

insolation and latitudinal forcing (low versus high latitude) – with a focus on the 2.3-1.2 Ma interval – which has not yet

been investigated in detail.

## 2 Materials and Methods

### 2.1 Setting and Chronology

ODP Site 967 is located on the northern flank of Eratosthenes seamount in the Levantine Basin at 34°04′N and 32°43′E (Fig.

1). Sediments from this site were recovered from a water depth of 2,554 m during ODP Leg 160. Dust fluxes to ODP Site

967 are sourced from Algeria, Libya, and western Egypt from 21-30°N latitudes; this dust is mainly transported during

boreal late winter and spring (Larrasoaña et al., 2003; Trauth et al., 2009). Fluvial sediment fluxes from the River Nile into

the eastern Mediterranean peak from June to October when discharge is high under the influence of monsoon-related Nile

catchment precipitation (Williams et al., 2006).

95   We use the composite depth splice and chronology described by Grant et al. (2017; 2022). The ODP Site 967 chronology

is based on alignment of peaks in principal component 2 (PC2; a proxy associated with monsoon run-off and Mediterranean



sapropel deposition, based on principal component analysis of XRF-CS data) to precession minima with zero phase lag, from 0.161 to 3.09 Ma, while ages from 0 to 0.161 Ma are constrained by radiometrically-based ages of sapropels and tephra layers (Grant et al., 2016; 2017).

## 2.2 XRF-CS and calibration assessment

The equipment settings used for XRF-CS measurements are described by Grant et al. (2017). In short, an Avaatech XRF core scanner was used to measure 90 m of core material at 1-cm intervals with energy settings of 10 and 50 kV, which produced intensity data (counts) for 11 target elements; i.e., Al, Si, S, K, Ca, Ti, Mn, Fe, Sr, Zr, and Ba. These intensity data were then converted into concentrations using the multivariate log-ratio calibration (MLC) approach of Weltje et al. (2015).

The calibration data set of Grant et al. (2017) was based on 40 ED-XRF analyses of bulk sediment powder samples using a PANalytical Epsilon3 XL instrument. Instead, we here use the 1060 discrete samples from ODP Site 967 measured by Konijnendijk et al. (2014; 2015) as calibration samples. These glass-bead samples were prepared by melting 600 mg of sediment with 6000 mg of lithium tetraborate, after which WD-XRF was performed with a Philips PW 2400 X-ray spectrometer. Both Grant et al. (2017) and Konijnendijk et al. (2014; 2015) reported analytical precisions of the ED-XRF and WD-XRF analyses better than 2% for Al and Ti, but comparison is ambiguous because different standard samples were used (MAG-1 and ISE 921, respectively). Yet, as noted before, glass bead WD-XRF is established as a more accurate and precise method than sediment powder ED-XRF (Zahn, 2005).

The MLC conversion was implemented using the AvaaXelerate software (Bloemsma, 2015; Weltje et al., 2015) that minimizes the impact of down-core physical property changes and parameterizes non-linear matrix effects. Within the software, we set the sample tolerance (i.e., the allowed maximum distance between a calibration sample and the XRF-CS data) to ±15 mm to allow for minor depth mismatches between calibration samples and XRF-CS measurement depths. Subsequent calibration was performed with all 1060 samples of Konijnendijk et al. (2014) and subsets thereof: (1) three evenly spaced subsets of 10%, 5%, and 2% of the samples (i.e., 106, 53, and 22 samples, respectively), and (2) two subsets (i.e., 53 and 22 samples) obtained using statistically selected sample positions with the automated calibration sample selection option in the AvaaXelerate software. This sample selection is based on the multivariate geometry of the XRF-CS intensities (Weltje et al., 2005) and provides the minimum of calibration samples based on the variance and number of elements in the XRF-CS dataset (Bloemsma, 2015). Conversion of intensities into concentrations was possible for all target elements except for sulfur, which is known to be semi-quantitative for measurements performed on glass beads due to its partial loss during sample preparation.

The calibrated XRF-CS records were compared with the WD-XRF data of Konijnendijk et al. (2014) to test statistically the similarity between the (un)calibrated XRF-CS data to this reference record. We tested for equality of variance (F-test; $\alpha$ = 0.05) and mean (two-tailed tests: one-way analysis of variance (ANOVA), Student t-test, and non-parametric Mann-Whitney test; all at the level of $\alpha$ = 0.05). Moreover, we performed an ordinary least squares regression and calculated the Pearson product-moment correlation coefficient (r) to measure the linear correlation between the XRF-CS and reference





WD-XRF data. To do so, the depths of the XRF-CS data were resampled to the same sample intervals of the WD-XRF dataset through linear interpolation using Analyseries (Paillard et al., 1996).

## 3 Results and Discussion

### 3.1 Accuracy of Ti/Al time series from XRF-CS

XRF-CS Ti/Al records are shown in Fig. 2 prior to calibration (intensities in Fig. 2a and log-ratios of these intensities in Fig. 2b) and after calibration (Fig. 2c-i) versus the reference Ti/Al data obtained using WD-XRF on glass beads (Konijnendijk et al., 2014). Statistical test results (Table 1) for comparisons between XRF-CS data and reference data indicate that uncalibrated Ti/Al (counts/counts; Fig. 2a) ratios have no similarity at $\alpha = 0.05$ level to the WD-XRF results and relatively low correlation to these reference data. Likewise, statistical results for log-ratios of Ti/Al intensities also have no similarity and low correlation to the reference data (Table 1). The previous Grant et al. (2017) calibration is marginally better, with a slightly higher correlation but still with significantly ($\alpha = 0.05$) different values from the reference data (Table 1). The new calibrations have much higher linear correlation to the reference data (correlation coefficients of 0.60-0.74, depending on how many samples are used; Table 1), while the means of the XRF-CS data become statistically similar to the reference data for calibrations with 5% (n=53), 10% (n=106), and all (n=1060) calibration samples (Table 1). The calibration samples that were automatically selected using a clustering algorithm (Bloemsma, 2015) produce only somewhat better results when 22 calibration samples are considered (Fig. 2i versus Fig. 2g; Table 1) and not for 53 samples (Fig. 2h versus Fig. 2f; Table 1). This may be due to the fact that the calibration samples are selected automatically based on the whole XRF-CS dataset (i.e., all measured elements) and not specifically Ti and Al data.

Our results demonstrate that XRF-CS can be used to acquire accurate Ti/Al data, as long as an appropriate number of calibration samples is used (i.e., at least 53 samples here for Ti/Al; Table 1). Ti and Al are challenging elements to measure with XRF-CS, because they are relatively light elements and hence prone to sedimentary inhomogeneities due to the fact that the XRF signal originates from only the upper few µm of sediment (Potts et al., 1997). This may also explain the significant ($\alpha = 0.05$) difference in variance observed between all XRF-CS data and the reference data (Table 1). The reference data have the highest amplitude variability (to more positive values) at turbidite intervals (Konijnendijk et al., 2014), which typically contain larger grain sizes that may result in ambiguous data for Ti and Al, because of their shallow XRF signals and associated grain-size effect (i.e., larger grains are coated in smaller grains and thus shallow XRF signals from light elements preferentially record the small grain geochemistry). Importantly, all of our new XRF-CS calibrations (Fig. 2d-i) have similar cyclic variations to the WD-XRF reference data, which would likely lead to similar paleoenvironmental interpretations. Uncalibrated XRF-CS Ti/Al ratios and log-ratios (Fig. 2a, b), on the other hand, are seemingly unusable for paleoenvironmental purposes.

Mismatch between uncalibrated and reference Ti/Al values is likely due to differences in sample preparation, analytical sensitivity, and matrix effects for the methods used. Matrix effects result from influences of the fluorescence of other

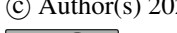



elements of interest in the sediment matrix by absorption or enhancement. For instance, Ca likely has a large impact on both Ti and Al because it is an effective absorber of Ti fluorescence and enhances Al fluorescence (Potts and Webb, 1992); it also has large down-core variations at ODP Site 967 (~2-33 weight %; Grant et al., 2022). Matrix effects and associated

analytical sensitivity (i.e., sensitivity to measure an element of interest by the XRF-scanner, being mainly dependent on X-Ray source, measurement geometry, instrument settings, and sample matrix) are approximately constant for WD-XRF analyses performed on well homogenized glass beads due to the 1:10 dilution and the low atomic weight of the lithium tetraborate flux (e.g., Konijnendijk et al., 2014; 2015). On the other hand, these parameters vary for XRF-CS measurements, which are performed directly on the split-core surface without sample preparation. Similarly, the quantitative ED-XRF used

to analyze sediment powders may have been less accurate for the same reason (Fig. 2c) because these powders have a more variable matrix than fused glass beads. Moreover, Zhan (2005) compared ED-XRF versus WD-XRF results and demonstrated that the data are generally comparable, although the WD-XRF technique is more precise and accurate. Hence, the substantial improvement of calibrated XRF-CS records (Fig. 2d-i; Table 1) is likely due to the use of WD-XRF calibration data from glass beads. The matrix effects and associated variable element sensitivities that clearly impact the

uncalibrated Ti/Al XRF-CS results (Fig. 2a, b) are appropriately accommodated by using these accurate calibration samples with the MLC method of Weltje et al. (2015).

The high-resolution Ti/Al record measured by XRF-CS, calibrated using all 1060 samples of Konijnendijk et al. (2014; 2015), was used to reconstruct past aridity and humidity variations over North Africa (see Section 3.2). The Ti/Al record of Lourens et al. (2001) independently validates our revised calibration (Fig. 3a).

**3.2 Arid/humid variability in North Africa over the last 3 Ma**

The ODP Site 967 Ti/Al record (this study; Grant et al., 2022) and wet-dry index of Grant et al. (2017) provide the longest, continuous, detailed representations of past North African climate variability (Fig. 3b), including information on humid period intensity compared to the dust proxy records. Ti/Al and the wet-dry index have mostly similar variability throughout the past 3 Ma, and the Ti/Al record confirms the timing and extent of "Green Sahara Periods" reported by Grant et al.

185   (2017).

The Ti/Al record indicates a clear increase in length and amplitude in arid intervals since approximately 1 Ma, with the highest Ti/Al values recorded from then on (Fig. 3a). This interval of enhanced intermittent aridity is coeval with the mid-Pleistocene transition (MPT; ~1.2-0.7 Ma; Clark et al., 2006; Chalk et al., 2017; Ford and Raymo, 2020; Berends et al., 2021) when ice ages intensified. Similar aridity increases have been observed in the Saharan dust supply record (Larrasoaña

et al., 2003) (Fig. 3c), and in records that capture dust inputs from subtropical West Africa (ODP 659; Tiedemann et al., 1994) (Fig. 3d) and the Arabian Peninsula and Horn of Africa (ODP 721/722; Bloemendal and deMenocal, 1989; deMenocal, 1995; 2004) (Fig. 3e). However, we also observe clear differences between ODP Site 967 Ti/Al and dust records from ODP Sites 659 and 721/722, which may be a result of the lower resolution of those records and chronological inconsistencies with the ODP Site 967 record. In addition, transportation effects by monsoon dynamics might play a role; the





ODP Sites 659 and 721/722 dust time series might not only record increased dust fluxes due to enhanced continental droughts but also due to stronger monsoon winds, which makes their interpretation more ambiguous (Trauth et al., 2009). Transportation impact by monsoon dynamics was likely minimal for the ODP Site 967 dust record (Trauth et al., 2009) and, thus, by extension, also for the ODP Site 967 Ti/Al record. Our Ti/Al record delivers an additional reliable and independent line of evidence that the most arid North African intervals after the MPT were unmatched in the preceding two million years.

The ODP Site 967 sapropel stratigraphy allows a relatively precise chronology throughout the Ti/Al record, by tuning of the sapropel geochemistry to precession minima (Grant et al., 2017; 2022). Organic-rich sapropel intervals in eastern Mediterranean sediments resulted from reduced deep-water ventilation and increased productivity during humid intervals in North Africa at precession minima (Rohling and Gieskes, 1989; Rohling et al., 2015). Regular Ba/Al peaks ("export-productivity"; e.g., De Lange et al., 2008) (Fig. 3f), thus, offer robust chronological constraints, paving the way for detailed

time-series analysis of the Ti/Al record. Importantly, the Ti/Al record was not used to tune the ODP Site 967 age model, and no lag was applied when tuning sapropel mid-points to precession minima (Grant et al., 2017). Previous studies indicate that precession minima can lead African monsoon maxima by about 3 kyr (Lourens et al., 1996; Ziegler et al., 2010; Konijnendijk et al., 2014). However, more recent work has shown that such a lag of the African monsoon to insolation only occurred after glacial terminations and principally after the MPT (Grant et al., 2016; 2017). Thus, before the MPT, and

especially during the Late Pliocene and Early Pleistocene, any lag was likely smaller or absent (Lourens et al., 2001; Grant et al., 2016; 2017).

Wavelet analysis and band-pass filtering highlight strong cyclic Ti/Al variability during the last 3-million-years (Fig. 4a-d). In general, we observe the expected enhanced North African humidity coupled to precession minima, obliquity maxima, and eccentricity maxima, and vice versa for North African aridity (Fig. 4b-d). The 100- and 400-kyr eccentricity bands are

stronger after the MPT, in line with other proxy records (e.g., deMenocal, 1995; Kaboth-Bahr et al., 2021). The eccentricity signal is much more apparent in Ti/Al than in the dust record from ODP Site 967 (see Larrasoaña et al., 2003). Eccentricity modulation of precession forcing strongly affects northward penetration of the African rainbelt into Saharan/Sahel watersheds, and thus the humidity recorded by Ti/Al. Moreover, eccentricity also impacts the El Niño-Southern Ocean oscillation, which includes the Walker circulation and thus affects pan-African climate (Kaboth-Bahr et al., 2021), offering

another mechanism for eccentricity imprint on Ti/Al. The eccentricity modulation of precession forcing is in phase with Ti/Al before the MPT, while the later phase relationship is more variable (Fig. 4d). This may be associated with a substantial climate system change at the MPT that produced a different North African aridity/humidity response on such long timescales. For instance, Trauth et al. (2009) suggested that the African climate response became more susceptible to remote high-latitude climate influences due to the crossing of a threshold ice volume during and after the MPT.

**3.3 Orbital and insolation forcing of ODP Site 967 Ti/Al variability**

The lag between obliquity forcing and the obliquity signal in the Ti/Al record holds information on low- versus high-latitude controls on North African aridity/humidity. The high-latitude ice-sheet response time to obliquity has been approximated to



be ~6.5 kyr (Lisiecki and Raymo, 2005; Lourens et al., 2010). Hence, a similar lag in North African aridity/humidity would provide evidence for high-latitude control, while a low-latitude control would instigate a much smaller lag. The relatively

small lead (~2 kyr) of obliquity relative to Ti/Al over the 3-1 Ma interval, thus, suggests that high-latitude climate had a relatively limited impact on monsoon activity in North Africa into the MPT (Fig. 4c). This holds true even if a relatively large (and unlikely; see discussion above) lag of ~3 kyr is assumed for most of the Pleistocene, because Ti/Al then still lags obliquity considerably less than the ~6.5 kyr ice-volume response to obliquity (Lisiecki and Raymo, 2005; Lourens et al., 2010). These results, therefore, point to an obliquity control on the low latitudes that affected North African aridity/humidity

to at least the MPT, which is consistent with data from a high-resolution coupled general circulation model (Bosmans et al., 2015a).

Climate model results of Bosmans et al. (2015a; 2015b) also indicate that the cross-equatorial insolation gradient provides a mechanism for the relatively large obliquity signal originating from low latitudes, which influences North African monsoon variability. The June 21st insolation at 65°N is often used to tune North African climate variability (Lourens et al.,

1996; Ziegler et al., 2010; Konijnendijk et al., 2014), which presumes a high-latitude control of North African climate. Our results and those of Bosmans et al. (2015a; 2015b) suggest that variability in the summer inter-tropical insolation gradient (SITIG), i.e., the difference in June 21st insolation between 23°N and 23°S (Lourens and Reichart, 1996), in principle provides a better tuning target curve for North African climate records (Fig. 4e, f).

SITIG and African humidity/aridity (Fig. 4e, g) have a high cross-correlation, which indicates similar amplitude

variability (and similar timing, but this is inherent to the age model construction). This high correlation is especially visible for insolation and the combined precession + obliquity signals in African humidity/aridity (Fig. 4f, g) from ~3 to 1.2 Ma. This is consistent with earlier research on the 3.2-2.3 Ma interval (De Boer et al., 2021; Lourens et al., 2001) and we show here that the African climate system operated in a similar manner until at least 1.2 Ma (Fig. 4g). During and after the MPT this linear correlation persists, but it is clearly diminished (i.e., with considerably lower correlation r, shifting from -0.9 to

about -0.45; Fig. 4g). This suggests that changes in the amplitude of Ti/Al during and after the MPT responded more variably ("noisy") to insolation changes, with a larger lag (~3 kyr; Konijnendijk et al., 2014). This implies a substantial global climate system change at the MPT that caused a different North African aridity/humidity response to insolation.

We observe North African aridity/humidity changes in parallel with changes in the ice age cycles at the MPT (i.e., ice ages intensified, lengthened from ~40 kyr to ~100 kyr, and became distinctly asymmetrical; Clark et al., 2006). Such large

cryosphere changes are represented by sea-level records that reflect global-scale ice sheet melt/growth during (de)glaciation. A 5.3-million-year record of relative sea-level changes at the Strait of Gibraltar ($RSL_{Gib}$) (Rohling et al., 2014) has similar chronologic constraints to our ODP Site 967 Ti/Al record; both are from the Mediterranean Sea and are constrained by sapropel chronologies (Wang et al., 2010; Rohling et al., 2014), which facilitates comparison (Fig. 5b). Although the Mediterranean sea-level record seems biased toward high values prior to ~1.2 Ma, its timing structure of relative variability

is consistent with that of other long-term sea-level reconstructions (Fig. 5a; Rohling et al., 2021). Correlation between $RSL_{Gib}$ and ODP Site 967 Ti/Al is low at the Plio-Pleistocene boundary and increases toward the MPT, after which higher

correlation is maintained (Fig. 5c). The observed coherence of these records points to an association of increased aridity (higher Ti/Al) with intensified, lengthened, and more asymmetric glaciations (and thus also lower sea level) from ~1.4 Ma to present, when sea-level changes started to frequently dip below ~65 meters (Fig. 5a). These results corroborate the data of

Ziegler et al. (2010) in which repeatedly suppressed African monsoon intensities coincided with North Atlantic cold intervals during the last 350 ka. This provides additional evidence that African climate became more sensitive to remote high-latitude climate influences around the MPT, so that the African monsoon response to (low-latitude) insolation became more non-linear after the MPT.

## 4 Conclusions

We investigate here XRF-CS calibration of bulk sediment Ti/Al from ODP Site 967 and use a well calibrated Ti/Al record to analyze responses and forcing of North African aridity/humidity in detail over the past 3 Ma, with a focus on the 2.3-1.2 Ma time interval.

Uncalibrated intensity ratios from XRF-CS can deviate significantly from ratios of the same elements measured by established geochemical methods using discrete samples, due to matrix effects and associated variable element measurement

sensitivities. Calibration using the multivariate log-ratio approach of Weltje et al. (2015), which estimates relative matrix effects and element sensitivity with selected calibration samples, efficiently corrects for this. We show that highly accurate calibration measurements are essential for proper calibration of these XRF-CS data and that calibration improves with the number of calibration samples. Here, at least 53 samples were necessary for proper calibration of the Ti/Al ratio. Ti/Al is a suitable proxy for tracing sedimentation processes in many other environments. Hence, our results are also relevant to

studies elsewhere that focus on Ti/Al from XRF-CS. Extending our results to other elements indicates that calibration may result in useful XRF-CS data for paleoenvironmental purposes, even if the initial intensity data for elements did not correspond with reference data.

Our ODP Site 967 Ti/Al record reveals a striking similarity in timing and amplitude between North African aridity/humidity and low-latitude insolation, especially during the 3 to 1.2 Ma interval. A small lead in obliquity to similar

frequencies in Ti/Al over that interval points to a low-latitude origin of this signal, which is consistent with climate model simulations. Our analyses imply that African climate became more sensitive to remote high-latitude climate when a threshold ice volume (sea level equivalent of below ~65 meters) was reached around the MPT.

## Data availability

The bead XRF elemental concentration data of Konijnendijk et al. (2014) can be found in the PANGAEA repository

(https://doi.org/10.1127/0078-0421/2014/0047). The calibrated XRF-scanning record of Grant et al. (2022) is essentially the same as the final calibrated XRF-scanning record presented here and is available in the PANGAEA repository





(https://doi.org/10.1594/PANGAEA.939929). We recommend to use that record for paleoenvironmental purposes. Note that our detailed analysis of the 2.3-1.2 Myr interval and extensive testing of the calibration approach is novel. The raw XRF-scanning data and results of the different calibration approaches used here are included in the Supplementary Material.

**Author contributions**

R.H. and K.M.G. conceptualized the project. RH conducted XRF-scanning calibration and data analysis, with consultation of all co-authors, and wrote the manuscript. All co-authors contributed to editing of the manuscript.

**Competing interests**

The authors declare that they have no conflict of interest.

**Acknowledgements**

R.H. was supported by the Netherlands Organisation for Scientific Research (NWO), with funding for the SCANALOGUE-project (ALWOP.2015.113) awarded to G.-J.R. This study was also carried out as part of the Netherlands Earth System Science Centre (NESSC; 024.002.001), supported by the Dutch Ministry of Education, Culture and Science (OCW). This work was also supported by the Australian Research Council (ARC) through grants DE1900100042 (K.M.G.), LE160100067 (ANZIC Legacy Grant; K.M.G. and A.P.R.), Australian Laureate Fellowship FL1201000050, grant DP200101157 (E.J.R.), and grant DP200100765 (A.P.R. and D.H.).

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





**Table and Caption**

**Table 1.** Statistical test results between XRF-CS data (uncalibrated and calibrated) and reference data. These data are also shown in Fig. 2. The 2nd column indicates the correlation coefficient r calculated between the datasets. The Y (Yes) and N (No) markers in the 3rd to 6th columns indicate whether the null hypotheses of equality of variance (3rd column) and means (4th to 6th column) could not (Y) or could (N) be rejected at the $\alpha = 0.05$ significance level. Hence, a Y indicates that the data are statistically similar.

| | Correlation r | Equality of variance (F-test) | One-way ANOVA | Student t-test | Non-parametric Mann-Whitney |
|---|---|---|---|---|---|
| Fig. 2a: Ti/Al intensities and Ti/Al WD-XRF | 0.34 | N | N | N | N |
| Fig. 2b: Ln(Ti/Al) intensities and Ln(Ti/Al) WD-XRF | 0.30 | N | N | N | N |
| Fig. 2c: Ti/Al Grant et al. 2017 calibr. (n=45) and Ti/Al WD-XRF | 0.39 | N | N | N | N |
| Fig. 2d: Ti/Al all calibr. samples (n=1060) and Ti/Al WD-XRF | 0.74 | N | Y | Y | Y |
| Fig. 2e: Ti/Al 10% calibr. samples (n=106) and Ti/Al WD-XRF | 0.68 | N | N | Y | Y |
| Fig. 2f: Ti/Al 5% calibr. samples (n=53) and Ti/Al WD-XRF | 0.68 | N | Y | Y | Y |
| Fig. 2g: Ti/Al 2% calibr. samples (n=22) and Ti/Al WD-XRF | 0.60 | N | N | N | N |
| Fig. 2h: Ti/Al AvaaXel. calibr. samples (n=53) and Ti/Al WD-XRF | 0.62 | N | N | N | N |
| Fig. 2i: Ti/Al AvaaXel. calibr. samples (n=22) and Ti/Al WD-XRF | 0.61 | N | N | N | N |



**Figures and Captions**

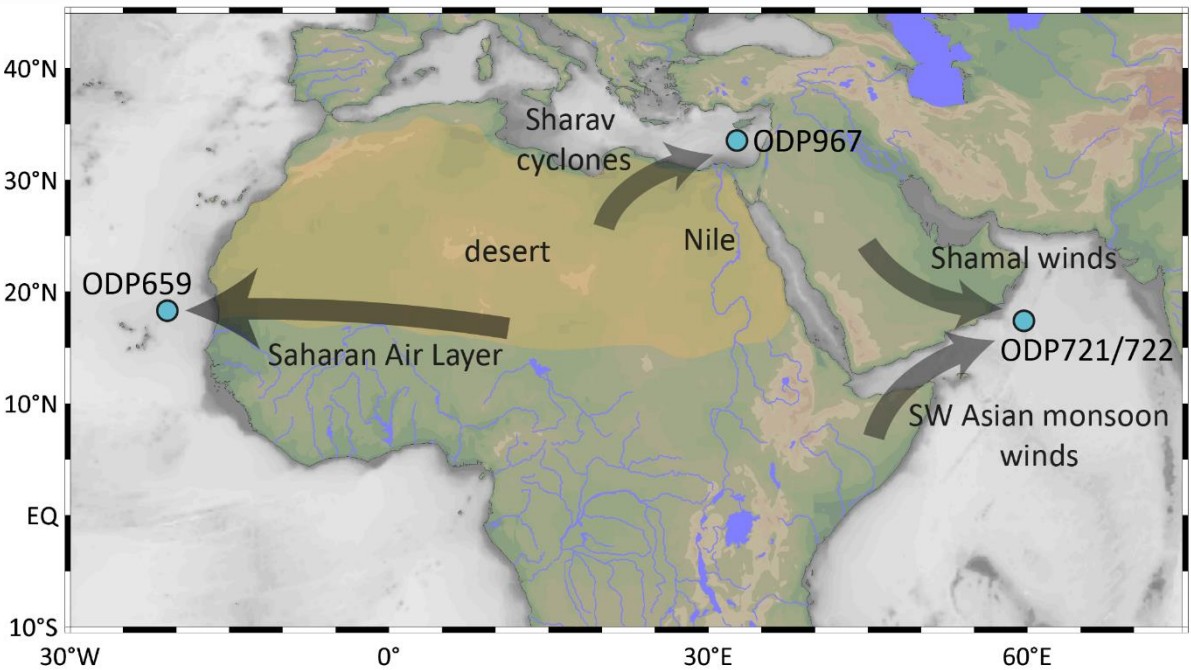

**Figure 1.** Location of ODP Site 967 in the eastern Mediterranean Sea, ODP Site 659 in the eastern Atlantic Ocean, and ODP Sites 721 and
722 in the Arabian Sea. The transparent black arrows indicate the main winds that transport dust to these sites (after Trauth et al., 2009).





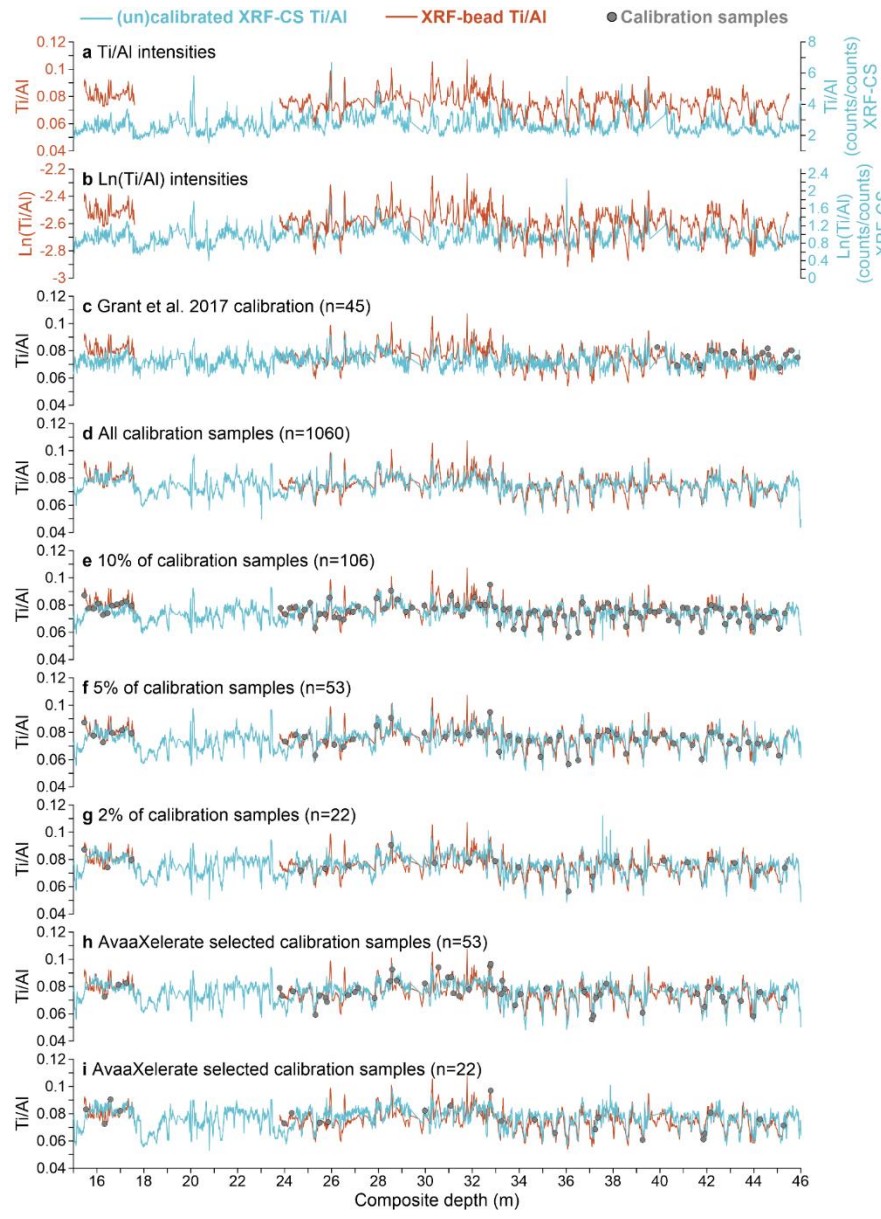

**Figure 2.** Uncalibrated and calibrated Ti/Al ratios (cyan) for different XRF-CS calibration strategies versus reference values obtained using WD-XRF on beads (orange; Konijnendijk et al., 2014; 2015). a) Ti/Al from XRF-CS intensities. b) Ln(Ti/Al) from XRF-CS intensities. c) Ti/Al using the calibration of Grant et al. (2017). d) Ti/Al using all 1060 calibration samples from WD-XRF on glass beads (Konijnendijk et al., 2014; 2015). e) Ti/Al using 10% of the samples from WD-XRF on glass beads. f) Ti/Al using 5% of the samples from WD-XRF on glass beads. g) Ti/Al using 2% of the samples from WD-XRF on glass beads. h) Ti/Al using the 53 samples selected by AvaaXelerate (Bloemsma, 2015). i) Ti/Al using the 22 samples selected by AvaaXelerate (Bloemsma, 2015). Gray circles represent data points used as calibration samples. Two high Ti/Al data points (0.13-0.14) were omitted from the XRF-bead record at 26 m for clarity but are incorporated in the statistical calculations. Statistical test results associated with these data are presented in Table 1.







**Figure 3.** The calibrated Ti/Al record from XRF-CS (this study; Grant et al., 2022; cyan) and companion records (orange). a) XRF-CS Ti/Al versus the WD-XRF values obtained by Lourens et al. (2001). b) XRF-CS Ti/Al versus the wet-dry index of Grant et al. (2017). c) XRF-CS Ti/Al versus Saharan dust supply (Larrasoaña et al., 2003). d) XRF-CS Ti/Al versus ODP Site 659 dust flux (Tiedemann et al., 1994). e) XRF-CS Ti/Al versus ODP Site 721/722 dust flux (DeMenocal, 1995, 2004). f) XRF-CS Ti/Al versus XRF-CS Ba/Al. Ti/Al and dust records in b-f are normalized (i.e. presented as z-scores by subtracting the average and dividing by the standard deviation over the same 3-million-year interval) to facilitate comparison.







**Figure 4.** Time-series analyses of the ODP Site 967 Ti/Al record. a) Wavelet analysis (Torrence and Compo, 1998) performed with a Morlet wavelet. The record was resampled to 1-kyr resolution before analysis. The cone of influence is indicated by the black line. Blue

contour lines indicate the p = 0.05 significance level. The cumulative signal strength is shown on the right. b) Filtered XRF-CS Ti/Al (cyan; 21±20% kyr) and precession (orange). c) Filtered XRF-CS Ti/Al (cyan; 41±20% kyr) and obliquity. d) Filtered XRF-CS Ti/Al (cyan; 100±20% plus 400±20% kyr) and eccentricity. e) XRF-CS Ti/Al versus the summer inter-tropical insolation gradient (SITIG). f) Filtered XRF-CS Ti/Al (precession plus obliquity bands) versus SITIG. g) Running correlation (401 kyr window) for profiles in (e; blue) and (f; orange); shadings indicate 95% confidence intervals. Wavelet analysis was done using the Past program (Hammer et al., 2001),

while band-pass filtering was performed with Analyseries (Paillard et al., 1996).





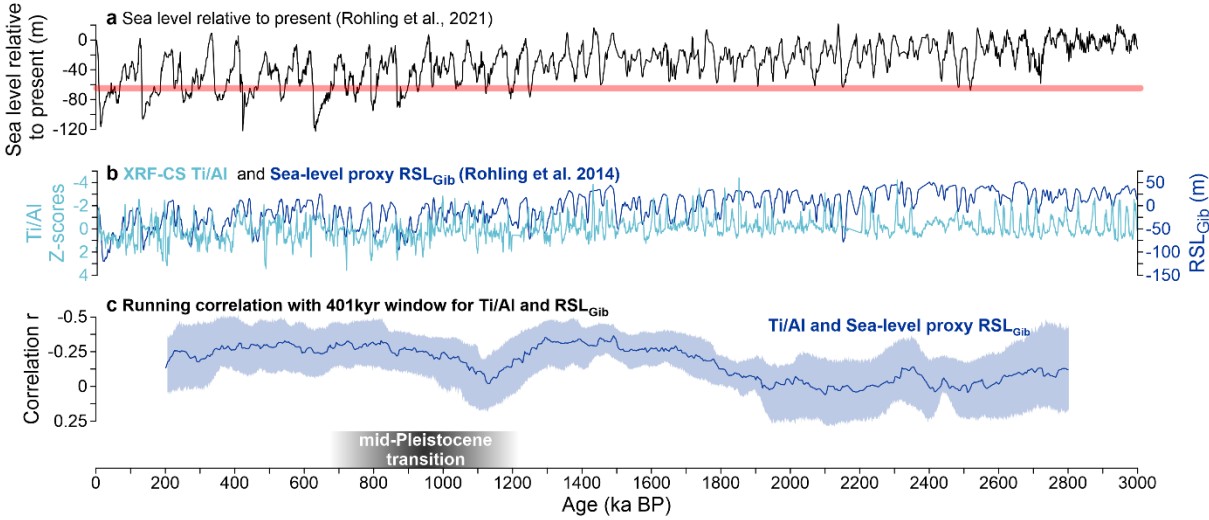

**Figure 5.** ODP Site 967 Ti/Al record and global ice-volume changes. a) Sea level relative to present (Rohling et al., 2021; black) reconstructed from deep-sea carbonate microfossil-based $\delta^{18}O$ of Westerhold et al. (2020). The red line marks the -65 meters sea level. b) ODP Site 967 Ti/Al (cyan) and relative sea-level change at Gibraltar (RSL$_{Gib}$) (Rohling et al. 2014; dark blue). Sapropel intervals are removed in this data set and data accordingly interpolated. b) Running correlation (401 kyr window) for profiles in (b); shading indicates 95% confidence interval.