# Peer review of "Accurately calibrated XRF-CS record of Ti/Al reveals Early Pleistocene aridity/humidity variability over North Africa and its close relationship to low-latitude insolation"

_Climate of the Past, 2022_

## Author Comment (AC1)

**RC1:**

Hennekam et al. provide a new calibration of XRF-CS derived Ti/Al measurements from Mediterranean core ODP 967 – a key site for the study of Plio-Pleistocene Saharan climatic variability. This is an important record which provides additional evidences for the timing and intensity of wetter/drier periods in the Sahara and the potential global/orbital controls of these fluctuations. The article is well written, with little grammatical revision required to the main body of the text. I believe this article asks two key questions: 1) how best can non-destructive and destructive geochemical methods be combined to provide an accurate record of past climatic variability? And 2) what can this new record inform about the long-term orbital influences on Saharan climatic variability throughout the Early Pleistocene to Mid Pleistocene?

The major strength of this manuscript is that it offers a valuable method to mitigate loss of material though WD-XRF analysis by instead selecting fewer (1060) samples to calibrate a non-destructive XRF-CS record (8497 samples). This permits a higher resolution Ti/Al record to be produced. However, I have a few concerns with this section.

**Reply:** We thank the reviewer for their positive and constructive comments on our work. Below, we reply in detail to the comments of this reviewer.

I believe the authors would benefit from emphasising the novelty of their study more clearly. Currently, on the basis of the text, it does not seem entirely clear how this calibration and XRF-CS record differs from that of Grant et al. (2022). Did the authors obtain new Ti/Al measurements? Or did they use those of Grant et al. (2022)? Similarly, did Grant et al. (2022) use the same WD-XRF dataset (Konijnendijk et al. 2014, 2015) to calibrate their record? Is this study using the same data and method as Grant et al. (2022), and simply testing how many samples are needed for accurate calibration? The authors must make the last two paragraphs of the introduction (and the materials and methods section) much clearer so that readers can establish the data output of this study.

**Reply:** This study and the study of Grant et al. 2022 were executed in parallel and hence there is indeed overlap between the studies (i.e., similar calibration approaches) but also important differences that merit a separate publication. First, there is a misconception that proper calibration of XRF-scanning data is only necessary to quantify the geochemical data. We show that appropriate calibration also allows to significantly improve (i.e., making it consistent with other established geochemical methods) the capturing of down-core geochemical variability. This will also make it possible to revisit old intensity datasets that were deemed unusable and extract useful paleoenvironmental data. We think it is important to explicitly describe our methods, including appropriate statistical testing, in a paper (and not just a supplement), as it is pivotal knowledge for many studies to come. We will highlight (in the last two paragraphs of the Introduction, but also in the Abstract and Conclusions) more clearly that the misconception about quantified XRF-scanning exists and that accurate calibration can much improve the capturing of geochemical variability. Moreover, we will clarify, shortly, in the Introduction the differences with Grant et al. (2022) and we will add a paragraph within the Methods section that will describe similarities and differences between Grant et al. (2022) and this study in detail. The calibration set of Grant et al. (2022) also used the WD-XRF dataset of Konijnendijk et al. (2014, 2015). Second, Grant et al. (2022) consider the full 5 Myr scanning XRF records

from ODP967 in conjunction with other new proxy records from the same samples (stable isotopes and environmental magnetism), with a particular focus on the geochemical and lithological shift at 3.2 Ma, while we here focus on the 2.3-1.2 Myr interval. The available Ti/Al records (De Boer et al., 2021; Konijnendijk et al., 2014; 2015; Lourens et al., 2001) showed that the North African climate system seemed to behave differently after 1.2 Ma compared to before 2.3 Ma. Yet, that left a knowledge gap on the operation of the African climate system between 2.3-1.2 Ma, which we here for the first time address in detail. We will amend the Introduction to more clearly highlight this novelty.

The results table (Table 1). Instead of a Y or N value to indicate whether the null-hypotheses have been rejected, the authors should provide the P-value and test specific values. This could be included in supplementary material rather than the main text, but they must be accessible for researchers. Additionally, the authors need to account for the "multiple comparison problem" by adjusting the ð• ' Ž value

**Reply:** We will add relevant test results to Table 1, with a focus on the *p* values. To correct *p*-values for multiple comparisons, we will use the Bonferroni method (i.e., $p < = 0.05$ / #tests). We already applied this correction to our dataset and this will lead to one adjustment in Table 1 (see adjusted Table below). Specifically, the *p* value of the one-way ANOVA for Fig. 2e (10% calibration samples) is 0.0412 and hence this indicates that statistically, with the Bonferroni correction, the means of the data obtained with this XRF-scanning calibration and the WD-XRF data are not significantly different (= similar; a "Y" in the table). We will adjust the Methods section accordingly (i.e., the part that describes the statistical approach).

| | Correlation r | Equality of variance (F-test) | One-way ANOVA | Student t-test | Non-parametric Mann-Whitney |
|---|---|---|---|---|---|
| Fig. 2a: Ti/Al intensities and Ti/Al WD-XRF | 0.34 | N (<0.0001) | N (<0.0001) | N (<0.0001) | N (<0.0001) |
| Fig. 2b: Ln(Ti/Al) intensities and Ln(Ti/Al) WD-XRF | 0.30 | N (<0.0001) | N (<0.0001) | N (<0.0001) | N (<0.0001) |
| Fig. 2c: Ti/Al Grant et al. 2017 calibration (n=45) and Ti/Al WD-XRF | 0.39 | N (<0.0001) | N (<0.0001) | N (<0.0001) | N (<0.0001) |
| Fig. 2d: Ti/Al all calibration samples (n=1060) and Ti/Al WD-XRF | 0.74 | N (<0.0001) | Y (0.4383) | Y (0.4817) | Y (0.9299) |
| Fig. 2e: Ti/Al 10% calibration samples (n=106) and Ti/Al WD-XRF | 0.68 | N (<0.0001) | Y (0.0412) | Y (0.0701) | Y (0.4184) |
| Fig. 2f: Ti/Al 5% calibration samples (n=53) and Ti/Al WD-XRF | 0.68 | N (<0.0001) | Y (0.2573) | Y (0.2813) | Y (0.6464) |
| Fig. 2g: Ti/Al 2% calibration samples (n=22) and Ti/Al WD-XRF | 0.60 | N (<0.0001) | N (0.0077) | N (0.0101) | N (<0.0001) |
| Fig. 2h: Ti/Al AvaaXelerate calibration samples (n=53) and Ti/Al WD-XRF | 0.62 | N (<0.0001) | N (0.0006) | N (0.0010) | N (<0.0001) |
| Fig. 2i: Ti/Al AvaaXelerate calibration samples (n=22) and Ti/Al WD-XRF | 0.61 | N (<0.0001) | N (<0.0001) | N (<0.0001) | N (<0.0001) |

It is necessary for the authors to better explain why 53 samples are required for accurate calibration, and why, if this is sufficient, the 1060 sample calibration record is favoured for the subsequent discussion. I understand that it is necessary to reduce the number of samples to achieve the authors aims. However, I believe the justification for this amount is unclear as the test specific results have not been made available.

**Reply:** Indeed, 53 samples seem to be appropriate statistically, as the tests indicate similarity of the means between the XRF-scanning data and the XRF-bead data (we will provide this data in a new Table 1, see above) and a relatively high correlation coefficient r. However, this also indicates that the 1060 sample calibration performs best (i.e., highest *p* values of the tests and highest correlation r), which is why this calibration is favored for further use here. When such an extensive sample set would not be available already a 53 sample calibration set would be enough. We will clarify this in the Discussion.

For the high-resolution XRF-CS Ti/Al analysis and correlation to orbital records, I would like to first say that I am generally supportive of this analysis. The authors provide a detailed insight into the varying controls of orbital parameters on African wetter/drier periods. Unlike hematite dust transport, Ti/Al ratios provide a method to study the intensity of wetter/drier periods. Their statistical analysis and interpretation, that high-latitude forcing played an increasingly dominant role after the Mid Pleistocene Transition, appears reasonable and well argued. However, I believe this section needs further work and clarification/justification.

Firstly, the application of a 401 kyr window running correlation (long eccentricity band), based on the text, does not seem justified to the reader. Why was this running correlation window selected? The authors must explain why such a large window is necessary and crucial to their analysis and interpretations.

**Reply:** The 401-kyr window was chosen to obtain a relatively smooth running correlation record that focuses on long time-scale changes in variability. If the window is set to much shorter values (e.g., 100 kyr), then the smaller discrepancies between Ti/Al and insolation become more apparent, as fewer datapoints are involved. The ChangePoint statistics (see below), for instance, would probably indicate many more change points. With the 401-kyr window these statistics only focus on the largest changes in the record. We will justify and explain this in more detail in the Methods section of the revised version of the paper.

Secondly, as can be seen from the very well-made figures, the 95% confidence intervals (while they do represent extremes) are large and, considering this, there is some uncertainty when distinguishing the shift from low to high running correlation between >1.2 and <1.2 Ma. This is more of an issue for the correlation with sea-levels. Additionally, the claim for constant high correlation with sea-level after the MPT is not so clear; it appears that higher correlations exist from about 1.7 Ma, with an abrupt dip at 1.1 Ma, after which the high correlation returns. Perhaps the authors could perform a t-test of running correlation values between these two periods to test for significant differences?

**Reply:** Based on the comments of both reviewers (RC1 and RC2) and recent updates on sea-level proxy records, we have removed the running correlation (Fig. 5c in current manuscript) between Ti/Al and sea-level change at Gibraltar (RSL$_{Gib}$) from the revised version of the paper. The latest sea-level review produced by several co-authors involved in our study (Rohling et al., Submitted) shows that, compared to other sea-level proxy records, the RSL$_{Gib}$ deviates quite considerably before ~1.5 Ma. We therefore investigated the ODP967 Ti/Al running correlation with the recent Rohling et al. (2021) sea-level record based on deconvolution of deep-sea benthic foraminiferal $\delta^{18}O$ records, even though the latter age models might differ somewhat with our ODP967 record. However, we found that the resultant running correlation remained close to 0 within uncertainties. In light of this, and the concerns of both reviewers about the weak/variable correlation between Ti/Al and sea-level, we will now present a straightforward cross-correlation between sea-level and ODP967 Ti/Al values older/younger than 1.2 Ma (new Figure 5b – see below) and box-whisker plots of the same values (new Figure 5c).

Considering these new plots with our change-point and wavelet analyses, we believe that the evidence suggests a high latitudinal impact on North African climate around the MPT, despite a weak running correlation of Ti/Al and sea-level. For example, (1) wavelet analysis of

the Ti/Al record (Fig. 4a) shows a strengthening of wavelengths >100 kyr at the MPT, similar to high-latitude records; (2) Change point analysis highlights that indeed a statistical change occurs in Ti/Al at the MPT; (3) new Fig. 5b shows that large amplitude changes in both SL and Ti/Al share at least their timing, albeit nonlinearly; (4) The mean (t-test) and variance (F-test) is significantly different for both Ti/Al and sea level before and after MPT (new Fig. 5c; we will add these statistical results to the caption and text; all $p$ values are <0.0001). We will slightly adjust the end of the discussion to include these points, and we will remove the text about the running correlation.

[Figure]

Furthermore, both the correlation with insolation (is this SITIG, 65N, 35N or 15N? Please clarify on figures) and sea-levels timing may benefit from further investigation using ChangePoint analyses. If using the R statistical software package, this can be achieved with packages such as BCP or ChangePoint. This may result in slightly different ages identified for these changes, but combined with the current analysis, would add an additional line of support to the authors argument. In either case, I believe that, while there is a deal of statistical uncertainty, the authors analysis provides important information for understanding the orbital controls on Saharan wetter/drier periods throughout the Pleistocene.

**Reply:** We will clarify in the figures and captions that insolation means SITIG in this case. We thank the reviewer for this comment, because the ChangePoint analysis is indeed a great addition to the paper, which we will implement. Our new ChangePoint analysis on the correlation of Ti/Al and insolation identifies changes at: 317 ka, 1081 ka, 1404 ka for the mean and 286 ka, 1114 ka, and 1404 ka for the standard deviation. Indeed, most of these ages fall within or just prior to the MPT.

While I am supportive of their analysis, the authors may benefit from additional reference to various studies which describe the suppressive effects of glacial termination melt-water

discharge on low-latitude forcing during the Middle and Late Pleistocene, causing monsoon intensification to lag insolation (e.g., Marino et al., 2015; Menviel et al., 2021; Häuselmann et al., 2015; Böhme et al., 2015). While most of these studies are limited to the LIG or Holocene, this may provide an additional line of support for some of the authors arguments.

**Reply:** We will add the suggested literature (Böhm et al., 2015; Häuselmann et al., 2015, Marino et al., 2015; Menviel et al., 2021) – and we will slightly expand the text – at the end of the Discussion to further support the suppressive effects of glacial termination meltwater discharge on the North African monsoon system. This will aid in explaining the change in the phase relationship between (low-latitude) insolation and monsoon intensity during the late Pleistocene.

I recommend that this paper be published in Climate of the Past subject to the authors addressing the concerns and the few grammatical/technical notes below. I suggest minor revisions as 1) results of the statistical testing and consideration of the "multiple comparison problem" (this may have some impact on the results, but is difficult to estimate without seeing the test specific results); and 2) the interpretation/discussion needs further analysis and justification to support these arguments, and currently the novelty is not well emphasised. However, I believe that this work will make a valuable contribution once these concerns are addressed.

**Reply:** We again thank the reviewer for the constructive comments and hope that our proposed changes will take away any concerns.

Technical/grammatical notes:
Line 37-39: References. The authors may benefit from adding a few references to palaeoanthropological/archaeological outputs and discussions, that are not necessarily climatic research initiatives, to highlight the broader relevance of their work. (E.g., Potts et al. 2020; Groucutt et al. 2015)

**Reply:** We will add Potts et al. (2020) and Groucutt et al. (2015) to the literature already cited within these lines.

Line 58-86: The last two paragraphs of the introduction. I believe these paragraphs are, in short, saying "As WD-XRF is destructive, how many samples are required to accurately calibrate a non-destructive XRF-CS record?". The authors may benefit from revising these paragraphs to emphasise the aims of the manuscript more concisely (or perhaps directly). Maybe this is due to my unfamiliarity with the methods, but it took me a few attempts to work-out the novelty of this article, as Grant et al. (2022) is described as having conducted a very similar WD-XRF calibration of an XRF-CS record for the 5 Ma period of the core. The paragraphs must emphasise the novelty of this study more clearly.

**Reply:** As explained in our above responses, we will highlight the novelty more clearly within the Introduction, and will discuss the differences/similarities with Grant et al. (2022) within the Introduction and in more detail in the Methods section.

Line 172: There have been various comments that WD-XRF analysis is more precise/better established than other methods. Can the authors provide further quantification of this?

**Reply:** The statement about a more precise/accurate measurement of WD-XRF is especially focused on its comparison to ED-XRF. The WD-XRF technique reaches simply a much higher spectral resolution than ED-XRF, which results in better results. We will clarify this in the text.

Line 223-224. The authors may wish to add a comment on the work of Tzedakis et al. (2017). Nature, 542: 427-432.
**Reply:** We will add a comment on the work of Tzedakis et al. (2017), stating that at this time (around the MPT) an increase in the deglaciation energy threshold likely resulted in glacial cycles with lower frequency and higher amplitude.

Table 1. Please include the results of the statistical tests either here or in supplementary material.
**Reply:** As discussed above, we will.

Fig. 2 and caption. "XRF-bead". Perhaps change this to WD-XRF-bead for clarity?
**Reply:** We will adopt this change.

Fig. 3 may benefit from the addition of correlation coefficients of the XRF-CS Ti/Al record and the respective humidity/aridity records from ODP 967.
**Reply:** We will add this to the figure and shortly discuss it in the Discussion.

Fig. 4g. Please clarify if insolation is the SITIG, 65N, 35N or 15N.
**Reply:** We will clarify in caption and figure that this is SITIG.

Fig. 5c. The figure may benefit from a dashed line running horizontally from 0. This would allow the reader to track changes more easily in the correlation.
**Reply:** This comment is no longer relevant as we will adjust Fig. 5.

Figures. (not necessary). The cyan text may benefit from being a few shades darker.
**Reply:** We will adjust the cyan text to a darker blue (including the lines), as in the figure above.

**Literature**

Böhm, E., Lippold, J., Gutjahr, M., Frank, M., Blaser, P., Antz, B., Fohlmeister, J., Frank, N., Andersen, M., Deininger, M., 2015. Strong and deep Atlantic meridional overturning circulation during the last glacial cycle. Nature 517, 73-76.

De Boer, B., Peters, M., Lourens, L.J., 2021. The transient impact of the African monsoon on Plio-Pleistocene Mediterranean sediments. Clim Past, 331-344.

Grant, K.M., Amarathunga, U., Amies, J.D., Hu, P., Qian, Y., Penny, T., Rodriguez-Sanz, L., Zhao, X., Heslop, D., Liebrand, D., Hennekam, R., Westerhold, T., Gilmore, S., Lourens, L.J., Roberts, A.J., Rohling, E.J., 2022. Organic carbon burial in Mediterranean sapropels intensified during Green Sahara Periods since 3.2 Myr ago. Communications Earth & Environment 3, 1-9.

Groucutt, H.S., Petraglia, M.D., Bailey, G., Scerri, E.M., Parton, A., Clark-Balzan, L., Jennings, R.P., Lewis, L., Blinkhorn, J., Drake, N.A., 2015. Rethinking the dispersal of Homo sapiens out of Africa. Evolutionary Anthropology: Issues, News, and Reviews 24, 149-164.

Häuselmann, A.D., Fleitmann, D., Cheng, H., Tabersky, D., Günther, D., Edwards, R.L., 2015. Timing and nature of the penultimate deglaciation in a high alpine stalagmite from Switzerland. Quaternary Science Reviews 126, 264-275.

Konijnendijk, T., Ziegler, M., Lourens, L., 2015. On the timing and forcing mechanisms of late Pleistocene glacial terminations: insights from a new high-resolution benthic stable oxygen isotope record of the eastern Mediterranean. Quaternary Science Reviews 129, 308-320.

Konijnendijk, T.Y.M., Ziegler, M., Lourens, L.J., 2014. Chronological constraints on Pleistocene sapropel depositions from high-resolution geochemical records of ODP Sites 967 and 968. Newsletters on Stratigraphy 47, 263-282.

Lourens, L.J., Wehausen, R., Brumsack, H.J., 2001. Geological constraints on tidal dissipation and dynamical ellipticity of the Earth over the past three million years. Nature 409, 1029-1033.

Marino, G., Rohling, E.J., Rodriguez-Sanz, L., Grant, K.M., Heslop, D., Roberts, A.P., Stanford, J.D., Yu, J., 2015. Bipolar seesaw control on last interglacial sea level. Nature 522, 197-201.

Menviel, L., Govin, A., Avenas, A., Meissner, K.J., Grant, K.M., Tzedakis, P.C., 2021. Drivers of the evolution and amplitude of African Humid Periods. Communications Earth & Environment 2, 1-11.

Potts, R., Dommain, R., Moerman, J.W., Behrensmeyer, A.K., Deino, A.L., Riedl, S., Beverly, E.J., Brown, E.T., Deocampo, D., Kinyanjui, R., 2020. Increased ecological resource variability during a critical transition in hominin evolution. Science advances 6, eabc8975.

Rohling, E.J., Yu, J., Heslop, D., Foster, G.L., Opdyke, B., Roberts, A.P., 2021. Sea level and deep-sea temperature reconstructions suggest quasi-stable states and critical transitions over the past 40 million years. Science Advances 7, eabf5326.

Rohling, E.J., Foster, G.L., Gernon, T.M., Grant, K.M., Heslop, D., Hibbert, F.D., Roberts, A.P., Yu, J., Submitted. Comparison and synthesis of sea-level and deep-sea temperature variations over the past 40 million years. Reviews of Geophysics. pre-publication version: https://www.essoar.org/doi/abs/10.1002/essoar.10510904.1

Tzedakis, P., Crucifix, M., Mitsui, T., Wolff, E.W., 2017. A simple rule to determine which insolation cycles lead to interglacials. Nature 542, 427-432.

---

## Author Comment (AC2)

**RC2:**

**General comments**

The study by Hennekam et al. has two major objectives: (1) investigate how to reliably calibrate core-scanner elemental records using the example of core ODP 967 from the eastern Mediterranean Sea, and (2) discuss changes in North African humidity and aridity over the last 3 Ma (with a special focus on the Mid-Pleistocene Transition (MPT), ~1.2-0.7 Ma), and their drivers (orbital parameters, insolation, ice volume). First, the authors test various numbers of WD-XRF calibration samples and two ways of selecting calibration samples, and discuss how much the calibrated core-scanner Ti/Al record compares statistically to the reference Ti/Al record. Then, the authors discuss calibrated Ti/Al changes in terms of North African aridity changes over the last 3 Ma, in agreement with available high-resolution records. Aridity over North Africa was particularly enhanced after the MPT. They confirm the strong control of orbital parameters (precession, obliquity, eccentricity) on North African humidity. Whereas low-latitude forcing dominates between 3 and 1.2 Ma, North African climate became more sensitive to high-latitude climate forcing when global ice volume increased during the MPT.

The manuscript is concise, well written, easy to read. Figures are clear and well explained. The methodology is sound and generally clear. However, I have two major concerns about the manuscript. First, I find the two parts appear to be rather disconnected one from another. It almost gives the impression two small studies have been merged together to build a manuscript.
**Reply:** We thank the reviewer for their constructive comments. We understand that it might seem that the manuscript consist of two parts, but we believe that our findings in both parts are closely linked and sufficiently important to merit publication in Climate of the Past. Below, we reply in detail to the comments of this reviewer and we propose changes that will ensure that all parts are interconnected.

Second, I find it difficult to identify the new information this study brings in comparison to previous studies. It is stated (lines 292-293) :"our detailed analysis of the 2.3-1.2 Myr interval and extensive testing of the calibration approach is novel." (NB: Is the Data availability section the right place to make such a statement?) However, even if I find the calibration testing exercise interesting (though frustrating by lack of more detailed discussion), I wonder to which extent it is needed for the interpretation of the Ti/Al record (see specific comment 1 below). Also, even if I am not an expert of North African climate over these time scales, the manuscript gives the impression it confirms previous hypotheses on the control of North African humidity (rather than brings novel ideas). I also have the feeling the study brings more insight on North African changes during the MPT than between 2.3 and 1.2 Ma (as indicated in lines 85-86, and 293).
**Reply:** Indeed, our work does not yield entirely novel ideas on the control of the North African climate system, as several controlling mechanisms have been discussed before. However, we do provide insights in the main controlling factors of North African climate over the entire 3 Myr based on this Ti/Al record. Previously we did not know whether differences observed between records after the MPT (0-1.2 Ma) and long before the MPT (2.3-3.2 Ma) (De Boer et al., 2021; Konijnendijk et al., 2014; 2015; Lourens et al., 2001) occurred exactly at the MPT or within the 2.3-1.2 Myr interval. Hence, a knowledge gap existed concerning the impact of the

MPT on North African climate. Here we show, for the first time, that low latitude insolation primarily controls North African climate variability during the 3.2-1.2 Myr interval and that indeed a large change occured in North African climate during/after the MPT (although low-latitude insolation remains very important). We will more clearly highlight this novelty within the Introduction.

Concerning the calibration testing, there is a misconception that proper calibration of XRF-scanning data is only necessary to quantify the geochemical data. We show that appropriate calibration significantly improves (i.e., making it consistent with other established geochemical methods) the down-core geochemical variability. We find it important to emphasize this outcome in the paper, as it is useful and important for other XRF-scanning studies, especially those spanning long depth and/or time series. The Ti/Al ratios is perfect to showcase this, because both Ti and Al are major elements that seem easy to calibrate. But because they are elements that largely covary, their ratio is actually very challenging to correctly calibrate (e.g., this is shown by the intensity ratio variability deviating from the geochemical variability obtained with other methods). Therefore, we decided to merge these important findings on appropriate calibration with the above-mentioned paleoenvironmental interpretation. We will highlight (in the last two paragraphs of the Introduction, but also in the Abstract and Conclusions) more clearly that the misconception about quantified XRF-scanning exists and that accurate calibration can much improve geochemical variability. Moreover, we will explain why this Ti/Al record is specifically an excellent showcase. Doing so, will ensure that all parts of the paper are interconnected, with the calibration allowing the further interpretation. The statement in the Data availability section will be omitted. Moreover, we will add additional information on the calibration in the Discussion (see below for details).

In conclusion, I think this manuscript deserves publication in Climate of the Past, provided the authors are able to better highlight the added value of this study (compared to already published works), to better link the two parts of the manuscript (calibration exercise and interpretation of Ti/Al in terms of North African humidity) and to better highlight the usefulness of the calibration exercise for the study and the community, by further developing its discussion.

**Reply:** We again thank the reviewer for the constructive comments and hope that our proposed changes will take away any concerns.

**Specific comments**

Calibration testing exercise

I wonder to which extend the detailed exercise of comparing calibrations is really needed for the study, for 3 main reasons, which would all require additional discussion in the text.

(a) Why is the XRF calibration published by Grant et al. 2022 not included in the testing exercise? The study by Grant et al. 2022 is cited in lines 72-74. It uses 42 WD-XRF reference samples (cited as more accurate than ED-XRF samples). So why not include this calibration in the comparison?

**Reply:** We will clarify, shortly, in the Introduction the differences with Grant et al. (2022) and will add a paragraph within the Methods section that will describe similarities and differences between Grant et al. (2022) and this study in detail (see also comment and reply other referee).

*Moreover, we will add the Grant et al. (2022) calibrated data to Figure 3a to facilitate comparison and will provide appropriate statistics in the text.*

I had also been wondering how much the calibrated Ti/Al record published in Grant et al. 2022 differs from the Ti/Al record adopted in this study until I reached the Data availability section, where we can read (lines 290-291): "The calibrated XRF-scanning record of Grant et al. (2022) is essentially the same as the final calibrated XRF-scanning record presented here […]. We recommend to use that record for paleoenvironmental purposes." (NB: Is again the Data Availability section the right place for such a statement?) Above all, is a new calibrated Ti/Al ratio necessary here if it is the same as the one published by Grant et al. 2022?

**Reply:** *As mentioned above, we will omit this statement in the Data availability section. Moreover, we will highlight differences/similarities with Grant et al. (2022) in more detail.*

If the calibration exercise remains in the revised manuscript, I would advice to include the calibration by Grant et al. 2022 in the comparison, extend the discussion on how much calibrated Ti/Al records differ and clarify the ambiguous statement on the record recommended for paleoenvironmental purposes (the one by Grant et al. 2022 or the newly calibrated one with 1060 reference samples?), and for which reasons one record is preferred if they are essentially the same.

**Reply:** *Note that both this study, as well as Grant et al. (2022), use these 1060 WD-XRF samples (while Grant et al. 2022 also uses 42 extra samples to calibrate an older part of the record). Therefore, the Grant et al. (2022) calibration is almost identical as the one presented here and thus including this data in the calibration would be superfluous. We will address this in the paragraph that we will add to the Methods section, while adding the Grant et al. (2022) calibrated data to Figure 3a. This similarity between these studies exists because they were executed in parallel and therefore information on appropriate calibration could be shared at an early stage. However, this Climate of the Past paper is the appropriate platform to publish our important findings on XRF-scanning calibration.*

(b) The comparison between the different tests of calibration is based on five statistical tests comparing the calibrated core-scanner Ti/Al and the reference WD-XRF dataset (Table 1). I remain highly frustrated by the currently limited discussion (lines 141-147) on how many reference samples are recommended or suitable for the calibration, and which type of selection of reference samples should be preferred (even spacing or Xelerate automatic selection). I find it very difficult with this limited discussion to draw inferences for other calibration studies. In light of the exercise, what is the minimum recommended number of reference samples? (What about the recommendation by Weltje et al. 2015 (equation 21.15a) of having as number of calibration samples at least 3 times the number of elements to be calibrated?) How should reference samples rather be selected: evenly spaced, manually or automatically with Xelerate?

**Reply:** *We cannot recommend a specific number of samples to appropriately calibrate XRF-scanning data. This will depend on the sediment material (matrix) / site and length of the record etcetera. Moreover, this may also differ among the elements of interest. As such, our study provides a blueprint to carefully determine the minimal amount of samples to obtain accurate XRF-scanning data for a specific element / elemental ratio, including appropriate statistics. The*

Weltje et al. (2015) statement on having a number of calibration samples at least 3 times the number of elements to calibrate is more meant as a general guideline. Note that co-author R. Tjallingii was closely involved in the study of Weltje et al. (2015).

Indeed, I am quite surprised to see that even spaced samples seem to give a more robust calibration than automatically selected samples and would have liked to read a more extended discussion in lines 146-147. In summary, I would strongly advice to develop the discussion on the comparison of calibrations to make it more meaningful and useful to the community beyond the case study of core ODP 967.

**Reply:** This outcome came as a surprise to us as well, but this might (at least partially) be due to the fact that we focus on Ti/Al in this study. Other elemental data is actually improved using the data of the automatically selected samples (e.g. Ca, Fe, and Sr when 55 samples are used). As also stated below, we will provide the Xelerate figures with reference vs predicted concentrations as a supplement. As such, the community can appreciate these differences themselves. Moreover, we will shortly address this (i.e., add that other elements are calibrated better in our case with the automatically selected sampling + refer to the supplement) in the text of the discussion in lines 146-147.

Similarly, I would advice to add a direct comparison of the differently calibrated Ti/Al records and discuss their possible differences in the text. Indeed, at first sight from Figure 2, there do not seem to be major differences between the various Ti/Al records. Thus, the reader wonders why a detailed calibration exercise is included in the manuscript if all tested calibrations provide relatively similar calibrated log-ratios.

**Reply:** Figures with a direct comparison of the different calibrated Ti/Al records led to relatively unclear results (too much data in one plot). As such, that would not add to the message of the manuscript and therefore we decided to not add it to the revised manuscript. We believe that our new proposed Table 1 (now including the $p$ values from the statistical tests, while also including the correlation r values; see our reply to the comments of the other referee) provide the best statistically sound data to show and interpret differences between the various calibrations.

(c) So far, I thought that calibration of core scanner intensities was a requirement in provenance studies (where absolute values of elemental ratios are compared to the composition of source material) and a bonus in classical paleoenvironmental studies, as it is the case here. In my own experience, the calibration modifies the absolute values and amplitude of change of elemental ratios, but not so much their downcore variations. It does not seem to be the case here (Figure 2, mostly below 30 m) and I am curious to know why. Thus, I would also recommend to develop the discussion on how much the calibration modifies the uncalibrated Ti/Al record. I think it would make more convincing and better illustrate the statement of the necessity of the calibration for paleoenvironmental purposes (lines 158-159, 292). It would also reinforce the usefulness of having the exercise comparing the various calibrations within the manuscript and strengthen the link between the two "parts" of the manuscript.

**Reply:** We completely agree with the reviewer. Basically, this is the main reason why the calibration exercise should be part of the main article. There is a general misconception that

proper calibration of XRF-scanning data is only necessary to quantify the geochemical data, but we show that it modifies the variability (because of multivariate nature of the calibration and matrix + sensitivity corrections). This is largely already addressed in lines 160-176, but we will clarify this part of the Discussion to more appropriately emphasize this important outcome.

Finally, I think the information provided on the calibration would deserve clarification at two places. First, I would state more clearly in lines 122-124 that 10 elements are calibrated, give the name of calibrated elements and provide (as a supplement?) an illustration of the retained calibration for all elements (e.g. the Xelerate figure with reference vs. predicted concentrations). Second, I wonder how the authors managed to run the Xelerate software with 22 reference samples only for 10 calibrated elements, when I think the software requires as number of reference samples at least 3 times the number of elements (equation 21.15a in Weltje et al. 2015).

**Reply:** We will state in lines 122-124 that 10 elements are calibrated. Moreover, we will provide the Xelerate figures with reference vs predicted concentrations as a supplement. The Xelerate software also runs with less samples than "3 times the number of elements".

Changes in North African humidity
(a) As a non-expert on these long time scales, I would have liked to have more information on the chronologies and related age uncertainties. In particular:

What is an estimate of age uncertainties in core ODP 967 (lines 95-99)?
**Reply:** We here assume zero phase lag between precession minima and monsoon maxima. Other studies have adopted a 3-kyr lag between precession minima and monsoon maxima, based on data of the last glacial cycle which can radiocarbon dated (e.g., Konijnendijk et al., 2014). Yet, such a 2-3 kyr precession lag seems only appropriate after glacial terminations (Grant et al. 2016). Little or no lag seems appropriate for monsoon maxima not following glacial terminations and data prior to the MPT (Grant et al., 2016; 2017). As such, the exact phasing remains unknown, but the maximum uncertainty is likely ±3 kyr (see details in Grant et al., 2017). We will add a more detailed description on the expected age uncertainty after lines 95-99.

How were constructed the age models of sites ODP 659 and 721-722 (lines 192-194)? What is the related age uncertainty? What is an estimate of the age offset that is expected between the records of these sites and core ODP 967 for the period of interest?
**Reply:** The ODP659 has astronomically been tuned using a benthic $\delta^{18}O$ record (Tiedemann et al., 1994), while the age control of Sites 721/722 was obtained using an integrated oxygen isotopic (till 1.1 Ma), biostratigraphic, and magnetic polarity-based time scale (e.g., deMenocal, 1995). These articles do not specifically focus on the age uncertainties, but the uncertainty of the orbitally tuned parts are probably tied to the phasing taken relative to the orbital cycles. Hence, uncertainty is probably in the same range as for Site 967 (±3 kyr). Taking into account age uncertainties between two cores this could potentially amount to a maximum of 6-kyr off set. This is why we rather focus on showing the records alongside without calculation of any correlation statistics. Still, we assume that even with such offsets, similarities in amplitudinal

variability would have been clearly visible. We prefer to keep this part of the manuscript unchanged to not lose focus, as this age offset is not essential for our subsequent analyses.

How was estimated the small lead of 2 ka of obliquity over Ti/Al (line 230) and how does it compare to age uncertainties for the period?
**Reply:** The next sentence touches on this topic, stating that: "This holds true even if a relatively large (and unlikely) lag of ~3kyr is assumed …". In principle this refers to the maximum ±3 kyr age uncertainty. We rather keep the text unchanged, instead of calling it "age uncertainty", because this is a more accurate description. Still, like mentioned above, we will add information on the age uncertainty in the Methods.

(b) I would also have liked to read more detailed information (lines 235-239) on the climate model results (Bosmans et al. 2015a, b). Which type of simulations was run? Climate sensitivity experiments? What was exactly tested? Which results are observed?
**Reply:** Bosmans et al. (2015a, b) got their results using a fully coupled ocean-atmosphere general circulation model (EC-Earth), which is based on a weather forecast model with high resolution (~1.125°). This model allows a reliable representation of monsoonal rainfall and associated circulation patterns and they performed idealized experiments with different orbital parameters (minimum/maximum precession, minimum/maximum obliquity, and combinations thereof). Experiments were run for 100 years (with the first 50 years being a spin up) and primarily focus on changes in pressure gradients and moisture transport. A main conclusion that is relevant for our work is: "The EC-Earth experiments reveal that, instead of higher latitude mechanisms, increased moisture transport from both the northern and southern tropical Atlantic is responsible for the precession and obliquity signals in the North African monsoon. This increased moisture transport results from both increased insolation and an *increased tropical insolation gradient*." (Bosmans et al., 2015a). We will shortly add some details on the model in lines 235-239 (model type and experimental setup).

(c) I am not fully convinced by the values of the coefficients of correlation between Ti/Al and the Gibraltar relative sea level record (Figure 5c). Can we speak of a high correlation when the absolute value of the coefficient reach 0.3-0.4?
**Reply:** We agree with this reviewer. Based on the comments of both reviewers (RC1 and RC2) and recent updates on sea-level proxy records, we have removed the running correlation (Fig. 5c in current manuscript) between Ti/Al and sea-level change at Gibraltar (RSL$_{Gib}$) from the revised version of the paper. we will now present a straightforward cross-correlation between sea-level and ODP967 Ti/Al values older/younger than 1.2 Ma (new Figure 5b) and box-whisker plots of the same values (new Figure 5c). More details are given in our response to Reviewer 1 (including the new Figure 5).

Also in Figure 5 I think the method how "Sapropel intervals are removed in this data set and data accordingly interpolated" (line 475) should be explained.
**Reply:** See above comment. This comment is no longer relevant.

Technical comments

Line 27: I would write "the longest period and highest amplitude"
**Reply:** We will add "the" to the sentence.

Line 45-46: I would write "throughout the Pleistocene"
**Reply:** We will add "the" to the sentence.

Line 59: I would write "during 0-1.2 Ma and 2.3-3.2 Ma"
**Reply:** We will adopt this change.

Line 112: Please correct the reference "Zhan, 2005".
**Reply:** We will adopt this change.

Line 131: Please indicate which version of Analyseries has been used.
**Reply:** We will indicate this in the revised manuscript (Analyseries version 1.1.1).

Line 146-147: I would rather write "(i.e. all calibrated elements)".
**Reply:** Good point. We will adopt this change.

Line 459: Is the Ba/Al ratio shown in Figure 3f also calibrated? Please specify.
**Reply:** Yes, we will specify this in the caption in the revised manuscript.

**Literature**
Bosmans, J., Drijfhout, S., Tuenter, E., Hilgen, F., Lourens, L., 2015a. Response of the North African summer monsoon to precession and obliquity forcings in the EC-Earth GCM. Climate dynamics 44, 279-297.
Bosmans, J.H.C., Drijfhout, S.S., Tuenter, E., Hilgen, F.J., Lourens, L.J., Rohling, E.J., 2015b. Precession and obliquity forcing of the freshwater budget over the Mediterranean. Quaternary Science Reviews 123, 16-30.
De Boer, B., Peters, M., Lourens, L.J., 2021. The transient impact of the African monsoon on Plio-Pleistocene Mediterranean sediments. Clim Past, 331-344.
deMenocal, P.B., 1995. Plio-pleistocene African climate. Science 270, 53-59.
Grant, K.M., Grimm, R., Mikolajewicz, U., Marino, G., Ziegler, M., Rohling, E.J., 2016. The timing of Mediterranean sapropel deposition relative to insolation, sea-level and African monsoon changes. Quaternary Science Reviews 140, 125-141.
Grant, K.M., Rohling, E.J., Westerhold, T., Zabel, M., Heslop, D., Konijnendijk, T., Lourens, L., 2017. A 3 million year index for North African humidity/aridity and the implication of potential pan-African Humid periods. Quaternary Science Reviews 171, 100-118.
Grant, K.M., Amarathunga, U., Amies, J.D., Hu, P., Qian, Y., Penny, T., Rodriguez-Sanz, L., Zhao, X., Heslop, D., Liebrand, D., Hennekam, R., Westerhold, T., Gilmore, S., Lourens, L.J., Roberts, A.J., Rohling, E.J., 2022. Organic carbon burial in

Mediterranean sapropels intensified during Green Sahara Periods since 3.2 Myr ago. Communications Earth & Environment 3, 1-9.

Konijnendijk, T.Y.M., Ziegler, M., Lourens, L.J., 2014. Chronological constraints on Pleistocene sapropel depositions from high-resolution geochemical records of ODP Sites 967 and 968. Newsletters on Stratigraphy 47, 263-282.

Konijnendijk, T., Ziegler, M., Lourens, L., 2015. On the timing and forcing mechanisms of late Pleistocene glacial terminations: insights from a new high-resolution benthic stable oxygen isotope record of the eastern Mediterranean. Quaternary Science Reviews 129, 308-320.

Lourens, L.J., Wehausen, R., Brumsack, H.J., 2001. Geological constraints on tidal dissipation and dynamical ellipticity of the Earth over the past three million years. Nature 409, 1029-1033.

Tiedemann, R., Sarnthein, M., Shackleton, N.J., 1994. Astronomic timescale for the Pliocene Atlantic δ18O and dust flux records of Ocean Drilling Program Site 659. Paleoceanography 9, 619-638.

Weltje, G.J., Bloemsma, M.R., Tjallingii, R., Heslop, D., Röhl, U., Croudace, I.W., 2015. Prediction of Geochemical Composition from XRF Core Scanner Data: A New Multivariate Approach Including Automatic Selection of Calibration Samples and Quantification of Uncertainties, Micro-XRF Studies of Sediment Cores, pp. 507-534.

---

## Author Response (AR1)

**Author's response**

Dear Prof. Fleitmann,

Please find enclosed our revised manuscript "*Accurately calibrated XRF-CS record of Ti/Al reveals Early Pleistocene aridity/humidity variability over North Africa and its close relationship to low-latitude insolation*". We modified the manuscript following the constructive comments from the two reviewers (we thank them in the Acknowledgements). Copies of the manuscript with tracked changes, the revised manuscript (without tracked changes), and a file with additional supporting information (as requested by Reviewer 2) are all uploaded.

Based on the reviewer comments, we have amended the manuscript as follows (note: below we refer to lines in the manuscript *with* tracked changes):

- We emphasize the novelty of our study (lines 73 to 97) by making the last two paragraphs of the introduction clearer (RC1; RC2; Editor comments). We now also clearly state the connection between the more methodological (i.e., calibration testing) and paleoceanographic parts of the study (RC2 comment) (lines 73 to 79).
- We clarify briefly in the Introduction the differences/similarities with Grant et al. (2022) (lines 94-97), and in more detail in the Methods (lines 149-153). Moreover, we added the Grant et al. (2022) calibrated data to Figure 3a to facilitate comparison and adjusted the text accordingly (lines 205-206). We have removed the statement on the novelty of our study from the data availability section. (RC2 comment)
- We have added a more detailed description on the expected age uncertainty (lines 110-114). (RC2 comment)
- We provide the Xelerate figures with reference vs predicted concentrations as a supplement and added to lines 139-140 that 10 named elements are calibrated. We now discuss briefly that other elements are potentially calibrated better with automatically selected sampling (lines 169-170). (RC2 comment).
- We have added *p*-values to Table 1. Moreover, in the caption, we now state that to correct *p*-values for multiple comparisons, we used the Bonferroni method. We added this to the Methods (lines 144-145). (RC1 comment)
- We clarify and emphasize in the Discussion that there is a general misconception that proper calibration of XRF-scanning data is only necessary to quantify geochemical data, but we show that it modifies the variability (because of the multivariate nature of the calibration and matrix + sensitivity corrections) (lines 197-201). (RC2 comment)
- We clarify why we use the 1060 sample calibration (lines 203-205). (RC1 comment)
- We briefly address details of the Bosmans et al. (2015a; b) model in lines 261-262. (RC2 comment)
- We applied change point analysis to our dataset (Figure 4g) and now refer to this in the text (lines 276-277). (RC1 comment)
- Based on comments from both reviewers (RC1 and RC2) and recent sea-level proxy record updates, we removed the running correlation (Fig. 5c in the initially submitted manuscript) between Ti/Al and sea-level change at Gibraltar (RSL$_{Gib}$) from the revised

paper. Instead, we now present a straightforward cross-correlation between sea-level and ODP967 Ti/Al values older/younger than 1.2 Ma and box-whisker plots of the same values (new Figure 5). Hence, we also adjusted the associated text in the caption and Discussion (lines 290-296). (RC1 and RC2 comments)

o We added the suggested literature (RC1) and slightly expanded the text at the end of the Discussion to further support the suppressive effects of glacial termination meltwater discharge on the North African monsoon system (lines 296-301). (RC1 comment)

o We briefly address why we chose a 401-kyr window in our running correlation (lines 538-540). (RC1 comment)

o We incorporated the minor/grammatical notes of the reviewers following our replies to reviewer comments (including the change in shade of blue lines in the figures).

We think that the current manuscript meets the high standards for publication in *Climate of the Past*.

On behalf of all co-authors, yours sincerely,

Rick Hennekam
*NIOZ – Royal Netherlands Institute for Sea Research*
*E: rick.hennekam@nioz.nl*